

# Transfer learning based approach for lung and colon cancer detection using local binary pattern features and explainable artificial intelligence (AI) techniques

Shtwai Alsubai

Department of Computer Science, College of Computer Engineering and Sciences, Prince Sattam bin Abdulaziz University, Al-Kharj, Saudi Arabia

## ABSTRACT

Cancer, a life-threatening disorder caused by genetic abnormalities and metabolic irregularities, is a substantial health danger, with lung and colon cancer being major contributors to death. Histopathological identification is critical in directing effective treatment regimens for these cancers. The earlier these disorders are identified, the lesser the risk of death. The use of machine learning and deep learning approaches has the potential to speed up cancer diagnosis processes by allowing researchers to analyse large patient databases quickly and affordably. This study introduces the Inception-ResNetV2 model with strategically incorporated local binary patterns (LBP) features to improve diagnostic accuracy for lung and colon cancer identification. The model is trained on histopathological images, and the integration of deep learning and texture-based features has demonstrated its exceptional performance with 99.98% accuracy. Importantly, the study employs explainable artificial intelligence (AI) through SHapley Additive exPlanations (SHAP) to unravel the complex inner workings of deep learning models, providing transparency in decision-making processes. This study highlights the potential to revolutionize cancer diagnosis in an era of more accurate and reliable medical assessments.

# INTRODUCTION

Cancer stands as one of the foremost reasons for fatalities worldwide. Cancer cells have self-growth, genomic instability, and a high propensity for metastasis. Colon cancer, also known as colorectal cancer, is a kind of cancer that appears in the colon or the rectum of the large intestine. Lung cancer, on the other hand, develops in the lungs and is distinguished by uncontrolled cell development. Lung cancer contributes to 18.4% of global cancer related fatalities and colon cancer constitutes 9.2% of such global fatalities (*Bray et al., 2018*; *Bermúdez et al., 2021*). The combined incidence of lung and colon cancer is about

Corresponding author
Shtwai Alsubai,
sa.alsubai@psau.edu.sa

17%. The prevalent occurrence of cancer cell dissemination between these two organs is notable in cases where early detection is absent (*Toğaçar, 2021*).

Many tests are performed, encompassing imaging procedures such as X-rays and CT scans, sputum cytology, and the extraction of tissue samples (biopsy) to identify cancer cells and eliminate other potential disorders. The assessment of biopsy slides by competent pathologists is critical for establishing the final conclusion (*Yeh et al., 2016*). This study primarily relies on histological scans to automatically identify colon and lung cancer cases. Histopathological pictures are commonly utilized by professionals for evaluation, and they are critical in finding patient prognosis. In the past, healthcare professionals were required to undergo an extensive procedure for cancer diagnosis, involving the examination of histopathological images. Nevertheless, with contemporary tools available, this process can now be accomplished more efficiently and with reduced time and effort (*Toğaçar, 2021*). Artificial intelligence (AI) has become increasingly popular for its ability to rapidly process data and make assessments (*Sakib et al., 2022*).

Most prior study publications examined utilizing deep learning to categorize colon and lung cancer pictures simultaneously. Few writers worked solely on lung cancer categorization and others focused solely on colon cancer classification. There are few studies devoted only to the categorization of colon cancer. For example, *Bukhari et al. (2020)* employed different versions of ResNet convolutional neural network architectures for colonic adenocarcinoma diagnosis. ResNet-50 had the best accuracy of 93.91%. *Hatuwal & Thapa (2020)* classified cancer images of lungs into three types and employed convolutional neural network (CNN) for lung cancer detection. Nishio et al. performed lung tissue classification using homology-based image processing to help lung cancer survivals (*Nishio et al., 2021*).

The authors extracted four feature sets for picture classification using two forms of domain transformations. The attributes of the two categories were then merged to provide the final categorization results and achieved a 96% accuracy (*Masud et al., 2021*). By employing deep neural network design, they classified colon and lung images. The authors achieved 96% and 97% accuracy in identifying colon and lung tumors respectively (*Mangal, Chaurasia & Khajanchi, 2020*). The authors conducted the histopathological image classification for detecting colon and lung tumors using Darknet-19. Subsequently, feature sets were employed, and two optimization methods were utilized to identify and eliminate ineffective features. *Toğaçar (2021)* performed lung and colon cancer classification utilizing efficient feature sets developed by separating unimportant features from the remainder of the set were then merged. and categorized. They achieved a total accuracy of 99.69% using an SVM classifier. *Talukder et al. (2022)* detected colon and lung cancer by applying machine learning models and ensemble learning for deep feature extraction. They used a histopathological image dataset (LC25000) for evaluating the model.

The primary goal of this research is to create a diagnostic aid system for medical imaging of the lungs and colon. In simpler terms, the aim is to construct an automated system capable of accurately identifying subtypes of colon and lung cancer from histopathological images utilizing transfer learning models. Additionally, the study seeks to illustrate that robust accuracy outcomes can be achieved through feature engineering. Deep learning models, on

the other hand, are black box whose function is incredibly difficult to understand because of the network's intricate architecture. This study used an explainable AI technique to interpret results. Feature engineering is critical in the medical and diagnostic fields because it helps them to recognise the importance and effect of each characteristic on the categorization and detection of cancer types. By considering all factors, this study offers the following contributions.

- This research presents an Inception-ResNetV2 model to precisely categorize colon and lung cancer in histopathological cancer images.
- The proposed approach leverages the distinctive local binary patterns (LBP) characteristics, leveraging their unique information encoding to achieve improved accuracy in addition to a significant enhancement in computational efficiency.
- The achieved results are elaborated through the application of the explainable AI method known as SHapley Additive exPlanations (SHAP), illustrating the individual contribution of each feature to the predictive analysis.

The following is the paper's structure: 'Related Work' performs a thorough evaluation of the relevant literature on colon and lung cancer detection. The materials and procedures used in this investigation are described in 'Proposed Approach'. 'Results and Discussions' focuses on the presentation of experimental findings. Finally, 'Conclusions' discusses the findings and offers future study directions.

## RELATED WORK

In this section, diverse methods for classifying cancer based on cell type are discussed. The accurate identification of ovarian cancer types is paramount for tailoring personalized treatment plans for patients. Over the past decade, numerous studies have sought to enhance the cancer screening process, aiming to detect cancer in its preclinical stages. However, manual image analysis by expert pathologists is not only time-consuming but also lacks consistency across different individuals. Recently, machine learning models have been extensively utilized in the initial screening and identification of colon and lung cancer. Various methods are recommended for extracting features from ultrasonic images and subsequently classifying them. This section further investigates several cutting-edge machine learning-based approaches designed for the detection of cancer.

*Alyafeai & Ghouti (2020)* designed a lightweight deep neural network for cancer detection and categorization. They achieved results 20 times quicker than the current top models, and the new technique is well-suited for mobile device deployment. In another work, *Chaunzwa et al. (2021)* developed a predictor based on CNN for diagnosing squamous cell carcinoma and adenocarcinoma in cancer patients. CNN has been verified utilizing non-small cell lung cancer patient data from early-stage afflicted individuals collected at Massachusetts General Hospital in real time.

*Qin et al. (2020)* emphasised the frequency of lung cancer and the need for noninvasive computer-assisted diagnostics. Deep learning, which is used for autonomous diagnosis, has difficulties extracting fine-grained characteristics from single modality pictures, spurring the development of a unique architecture that merges positron emission tomography

(PET) and computed tomography (CT) images utilising a multidimensional attention mechanism. The experimental findings showed an area under the receiver operating characteristic (ROC) curve of 0.92 and improved fine-grained feature extraction, showing that the suggested technique is useful for noninvasive lung cancer diagnosis. *Masood et al. (2018)* highlighted the global impact of pulmonary cancer and the role of computer-assisted diagnosis systems, particularly advanced by IoT and deep learning techniques. The proposed DFCNet model, integrating deep learning and metastasis information, demonstrates a promising 84.58% accuracy in classifying lung cancer stages, suggesting its potential to improve radiologists' efficiency in nodule detection.

*Bukhari et al. (2020)* discussed the importance of colonic cancer and the reliance on histological inspection, as well as new uses of convolutional neural networks (CNNs) for digital diagnosis. Their study evaluated deep learning's function in identifying colonic cancer through the processing of digitised pathology pictures, reaching considerable accuracies with ResNet-50 leading at 93.91%. The findings implied that computer-aided technology has the potential to help pathologists evaluate surgical specimens for malignancy identification. *Sirinukunwattana et al. (2016)* used histological scans to detect and categorize four nucleus types in colon tumors. Their suggested approach does not need nuclei segmentation and can categorize them with an F-measure as high as 80.2%. *Kuepper et al. (2016)* published an unlabelled classification approach and employed infrared spectral images and different stages of colon cancer in their investigation.

By evaluating histopathological cancer pictures, *Babu et al. (2018)* designed a classification algorithm using Random Forest (RF) to forecast the presence of colon cancer. They began by converting the RGB pictures to the HSV plane and then applied wavelet decomposition for feature extraction. By adjusting the image magnification, they attained a classification accuracy of 85.4%. *Mo et al. (2018)* employed a Faster R-CNN technique for colon cancer detection. *Urban et al. (2018)* applied deep CNN on colonoscopy images and achieved a 96% classification accuracy for identifying polyps. *Suresh & Mohan (2020)* introduced a diagnostic methodology for lung cancer utilizing CNN with feature learning focused on nodule regions of interest (ROI). To enhance the sample size, they employed generative adversarial networks (GANs) to generate additional CT scan images from the Lung Image Database Consortium (LIDC) and Infectious Disease Research Institute (IDRI) databases. Employing CNN-based classification techniques, they attained a peak classification accuracy of 93.9%.

*Shakeel, Burhanuddin & Desa (2022)* designed a technique to identify lung cancer utilizing CT scan images using an improved neural network for image segmentation. The authors applied image processing and deep learning approaches to detect colon and lung cancer from a dataset containing histopathological images and achieved an accuracy score of 96.33%. The previous studies underscore the challenges in cancer detection, emphasizing the limitations of manual analysis and the quest for efficient, automated solutions. The current work contributes by introducing an Inception-ResNetV2 model for the classification of colon and lung cancer in histopathological images, addressing the need for advanced diagnostic tools. Additionally, the utilization of local binary patterns (LBP) enhances accuracy and computational efficiency. The study further advances transparency

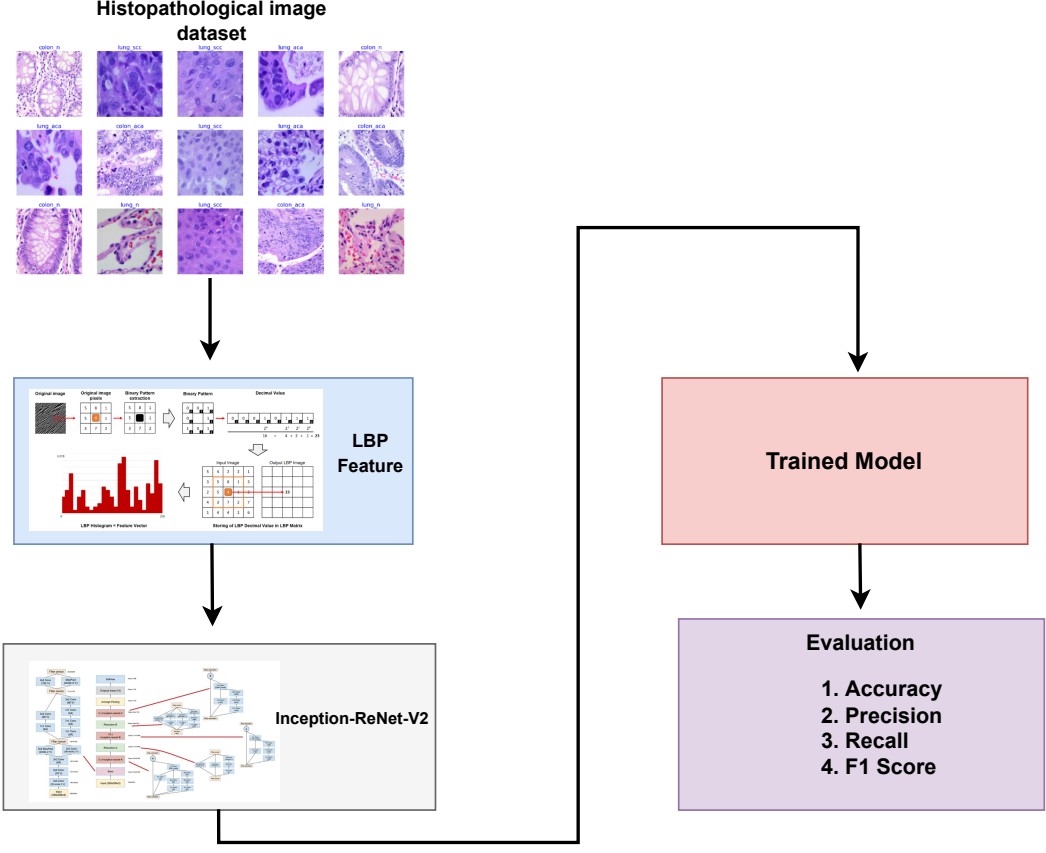

**Figure 1** Architectural diagram of the proposed methodology indicating the steps while arrows indicate the flow of steps.

in model interpretation through the application of explainable AI (SHAP), elucidating the contribution of each feature in predictive analysis, and bridging the gap between complex models and clinical understanding.

# PROPOSED APPROACH

This section discusses a detailed overview of the materials and methods utilized in classifying lung and colon cancer. It presents the specifics of the dataset employed, the transfer learning models employed for lung and colon cancer detection, and the assessment parameters used to evaluate the effectiveness of learning models. Figure 1 provides a graphical representation of the methodology employed in this study.

## Dataset

This study utilized a recently published histopathological image dataset called the LC25000 dataset (*Larxel, 2020*), which was introduced in 2020. It consists of 25,000 colored images representing five distinct types of lung and colon tissues. These variants are colon

adenocarcinoma, benign colonic tissue, lung adenocarcinoma, benign lung tissue, and lung squamous cell carcinoma.

- Colon adenocarcinoma, the most prevalent form of colon cancer, accounts for over 95% of all colon cancer cases. It arises from the development of a particular form of tissue proliferation known as adenoma in the large intestine, which subsequently progresses into cancer.
- Benign colonic tissue does not metastasize to different regions of the body.
- Lung adenocarcinoma, making up nearly 40% of all lung cancer cases and frequently affecting women, typically originates in glandular cells and extends towards the lung's alveoli.
- Benign lung tissue, do not metastasize to different regions of the body Although benign tumors are generally not life-threatening, they require surgical removal and evaluation through biopsy to confirm the absence of cancer.
- Lung squamous cell carcinoma is a form of small-cell cancer that emerges in the air passages or bronchi of the lungs. It stands as the second most prevalent form of lung cancer, constituting approximately 30% of all cases.

The authors collected 1,250 original photos of cancer tissues from pathology glass slides (250 for each tissue type). They applied image augmentation to rotate and flip the actual images under various situations to enrich the dataset, resulting in an enhanced dataset of 25,000 images (5,000 images for each class). Before applying the augmentation procedures, the source photos were cropped to 768 by 768 pixels to guarantee a square shape. All photos in the collection have been verified and comply with the Health Insurance Portability and Accountability Act (HIPAA) rules. Illustrations of images from the dataset are presented in Fig. 2.

## Local binary pattern

Local binary pattern (LBP) is a texture descriptor that is commonly used in computer vision for texture analysis and categorization. LBP works by examining the associations between pixels and their neighbours. Instead of processing the full image, LBP concentrates on specific areas of the image. A distinct neighbourhood of surrounding pixels is considered for each pixel in the picture. Based on its intensity levels in the neighbouring setup, each pixel is then compared to the central pixel. If the value of a neighbouring pixel is equal to or greater than the value of the centre pixel, it is assigned a value of one; otherwise, it is assigned a value of zero. This procedure produces a binary pattern, which is then transformed into a decimal number that represents the texture information in that particular area. A histogram of these LBP values is then created, providing a simple representation of the texture information in the image.

LBP is quite beneficial in a variety of applications like texture classification, face recognition, and object identification. Its resistance to changes in lighting conditions increases its value in circumstances where illumination varies. There are also several extensions and modifications of LBP, such as uniform LBP, rotation-invariant LBP, and other sorts of multi-scale and multi-block LBP, which broadens its application across a

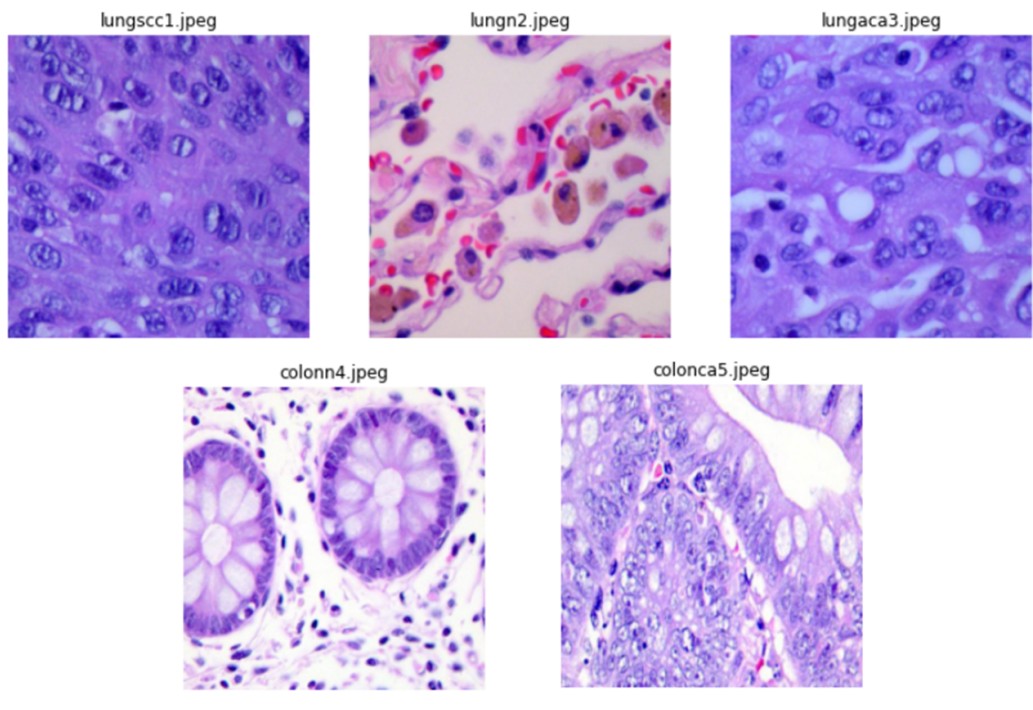

**Figure 2** **Sample images from dataset.**

wide range of circumstances. It is critical to recognise that, while LBP is a powerful texture descriptor, its usefulness is dependent on the individual application and used dataset.

## Deep and transfer learning models

CNNs are a key component of current deep learning, especially in computer vision (*Alturki et al., 2023*). These neural networks are built to replicate the human visual system, making them ideal for tasks such as object identification, image categorization and segmentation. CNNs are distinguished by their hierarchical architecture, which consists of numerous layers of convolutional, fully connected, pooling and convolutional layers. The network learns to recognize and extract essential information from input pictures in the convolutional layers by using filters that glide over the image, finding patterns and structures. Pooling layers help to minimize computing complexity by reducing the spatial dimensions of the data. CNNs are characterized by their ability to autonomously acquire knowledge and represent complicated hierarchical features from raw data, making them ideal for a variety of visual applications. These networks have transformed sectors like as medical imaging, self-driving vehicles, and face recognition, and their flexibility to multiple domains continues to drive artificial intelligence innovation and research.

VGG16, short for Visual Geometry Group 16, is used in computer vision and deep learning tasks. VGG16, created by the University of Oxford's Visual Geometry Group, is renowned for its ease of use and outstanding performance. It comprises 16 weight layers, 13 of which are convolutional and three of which are completely linked, allowing it to learn complicated characteristics from input pictures. VGG16's deep architecture

and compact 3 × 3 convolutional filters let it catch delicate features in pictures, making it particularly well-suited for image classification and object identification. Its simple structure has made it a popular candidate for transfer learning, where pre-trained VGG16 models are fine-tuned for multiple applications. VGG16's excellent accuracy, simplicity, and availability of pre-trained models have established it as a significant tool in the deep learning arsenal, particularly for image-based tasks such as image categorization, object recognition, and feature extraction.

MobileNet is a significant improvement in deep learning, aimed at overcoming the issues of implementing deep neural networks on resource-constrained devices like mobile phones and embedded systems. Google's MobileNet is distinguished by its efficiency and small architecture. It employs depth-wise separable convolutions, which considerably lower computational burden while keeping the capacity to capture critical characteristics. This architectural approach enables MobileNet to achieve high levels of accuracy in tasks like image categorization, object identification, and others, even on devices with limited processing power and memory. Its small weight makes it ideal for real-time applications like mobile image recognition and edge computing. MobileNet has evolved into an important tool in the field of computer vision, paving the way for the deployment of deep learning models in a broad range of edge devices, therefore contributing to improvements in domains such as mobile app development and the Internet of Things (IoT).

ResNet, short for residual network, is a ground-breaking deep learning architecture that has revolutionized the training of very deep neural networks (*Wang et al., 2019*). ResNet, developed by Microsoft Research, established the notion of residual connections, allowing the network to efficiently learn from and relay information across many layers. These leftover connections alleviate the vanishing gradient problem, allowing for the training of incredibly deep networks with hundreds of layers. ResNet architectures have become a cornerstone in picture classification, object recognition, and other computer vision applications, routinely producing robust results on benchmark datasets. ResNet's skip connections are a major breakthrough, allowing for the training of networks with remarkable depth, and they have since inspired the design of many other deep-learning models. ResNet's influence goes beyond computer vision research, affecting research in natural language processing and reinforcement learning, and it continues to be a fundamental architecture in the deep learning environment.

EfficientNetB4 is a deep learning model of the EfficientNet family that is noted for its remarkable efficiency and performance (*Zulfiqar, Bajwa & Mehmood, 2023*). These models were created by Google AI and are intended to maximize the balance between model size, accuracy, and computing efficiency. EfficientNetB4 is one of the bigger variations, providing greater accuracy without significantly increasing computing needs when compared to lesser ones. It accomplishes this by employing a compound scaling mechanism that simultaneously scales the model's depth, breadth, and resolution. This method makes EfficientNetB4 well-suited for a variety of computer vision tasks, such as picture categorization and object recognition, where it regularly produces cutting-edge results. The EfficientNet architecture has become a notorious alternative for transfer learning, allowing practitioners to fine-tune pre-trained models for specific applications

with reduced computational effort. It exemplifies the ongoing effort in deep learning to develop models that are both efficient and effective, making it a valuable asset for various applications in computer vision.

Google Research's Xception (*Salim et al., 2023*) is a deep learning architecture and it is known for its novel approach. The use of depthwise separable convolutions, which drastically reduce the parameters and computational complexity while retaining good accuracy, is the essential breakthrough of Xception. This design is based on the Inception model's concept of utilizing different filter sizes inside a single layer, but takes it a step further by employing depthwise separable convolutions across the network. Xception gets outstanding performance in picture classification and object detection tasks and is noted for its ability to generalize effectively even with less training data. This makes it useful for transfer learning and fine-tuning diverse computer vision tasks. Xception is a key step toward more efficient and scalable deep learning models by eliminating duplicate calculations and parameters. It impacted the construction of succeeding neural network designs and continues to be an important model in the fields of computer vision and deep learning.

Inception-ResNetV2 is a deep learning architecture that incorporates parts from the Inception and ResNet models (*Mujahid et al., 2022*). This hybrid architecture, developed by Google, is intended to use the characteristics of both networks in order to achieve high accuracy in picture categorization and other computer vision applications. Inception-ResNetV2 combines the Inception modules for efficient feature extraction with the ResNet residual connections to overcome the vanishing gradient problem and enable the training of very deep networks. The model is well-known for its remarkable performance on picture recognition tasks, with top scores on a variety of benchmark datasets. Inception-ResNetV2 is especially useful for applications requiring high accuracy and the capacity to interpret complicated visual input. It strikes a compromise between model size and computational performance, making it a flexible deep-learning tool. The inclusion of the ideas of Inception-ResNetV2 has affected the development of succeeding neural network designs, and it remains a noteworthy model in the field of computer vision and deep learning.

### Evaluation parameters

Multiple assessment measures are used in this work to evaluate the performance of classifiers. F1 score, accuracy, recall, and precision are among the metrics generated based on the values of false negatives (FN), true positives (TP), true negatives (TN), and false positives (FP).

Accuracy measures the overall accuracy of the model's predictions by dividing the number of correctly categorised samples (both positive and negative) by the total number of samples in the dataset.

$$Accuracy = \frac{TP + TN}{TP + TN + FP + FN}. \tag{1}$$

Precision is the proportion of accurately detected positive samples among all positive samples expected. It is calculated using the formula below.

$$Precision = \frac{TP}{TP + FP}.$$  (2)

Recall, also known as delicacy or true positive rate, measures a classifier's ability to accurately recognise positive samples inside a given class. The recall is determined using

$$Recall = \frac{TP}{TP + FN}.$$  (3)

The F1 score is used when there is a data class imbalance, integrating precision and recall into a single score. It is provided by

$$F1 - Score = 2 \times \frac{precision \times recall}{precision + recall}.$$  (4)

These assessment metrics aid in completely analysing classifier performance by accounting for several elements of their predictions, such as false negatives, false positives, true positives, and true negatives.

The Matthews correlation coefficient, commonly abbreviated as MCC, was introduced by Brian Matthews in 1975 (*Chicco & Jurman, 2020*). MCC serves as a statistical metric employed for model evaluation. Its primary function is to assess the disparity between predicted values and actual values, akin to the chi-square statistics utilized in a $2 \times 2$ contingency table.

MCC stands out as an optimal single-value classification metric that effectively summarizes the confusion matrix or error matrix (*Chicco & Jurman, 2010*). The confusion matrix comprises four entities: It is computed using the following formula:

$$MCC = \frac{TP \times TN - FP \times FN}{\sqrt{(TP + FP)(TP + FN)(TN + FP)(TN + FN)}}.$$  (5)

## RESULTS AND DISCUSSIONS

This study uses a variety of deep-learning and transfer-learning models to categorize histopathological cancer images. The dataset is divided into training and testing groups in a 70:30 ratio, which is a typical strategy used in many classification experiments to avoid overfitting. A range of assessment criteria including Accuracy, precision, recall and F1 score are used to evaluate model performance. All experiments are carried out in a Python environment with several libraries on a Dell PowerEdge T430 GPU with 2GB of RAM. This GPU is outfitted with twin Intel Xeon processors, each with eight cores running at 2.4 GHz, and 32 GB of DDR4 RAM.

### Results of the transfer learning models with full feature set

This section presents the results of transfer learning models with a full feature set of cancer image dataset. Table 1 shows the classification results of the models using all feature sets. The highest results have been achieved by the InceptionResNetV2 model with 98.96%

Table 1  Results of the models using full feature dataset.

| Model | Accuracy | Precision | Recall | F1 score |
|---|---|---|---|---|
| CNN | 84.65 | 86.86 | 86.70 | 86.78 |
| VGG16 | 80.65 | 82.57 | 81.99 | 81.59 |
| ResNET | 87.91 | 88.92 | 89.96 | 89.90 |
| EfficientNetB4 | 88.67 | 90.56 | 90.56 | 90.56 |
| MobileNet | 89.69 | 92.62 | 91.89 | 92.57 |
| Xception | 94.18 | 94.65 | 93.66 | 93.38 |
| InceptionResNetV2 | 95.96 | 94.65 | 93.43 | 93.98 |

Table 2  Classification results of the models using LBP features.

| Model | Accuracy | Precision | Recall | F1 score |
|---|---|---|---|---|
| CNN | 89.50 | 91.64 | 91.19 | 91.39 |
| VGG16 | 85.69 | 87.99 | 85.39 | 85.60 |
| ResNET | 94.25 | 95.37 | 95.68 | 95.73 |
| EfficientNetB4 | 95.86 | 97.17 | 97.32 | 97.24 |
| MobileNet | 96.43 | 98.02 | 98.19 | 98.14 |
| Xception | 98.37 | 99.48 | 98.15 | 98.97 |
| InceptionResNetV2 | 99.98 | 99.99 | 99.99 | 99.99 |

accuracy, 94.65% precision, and 93.98% F1 score. The Xception model has also shown remarkable results with 94.18% accuracy, 94.65% precision, 93.38% F1 score and the highest recall score of 93.66%. The MobileNet and EfficientNetB4 have shown almost similar results with 89.69% and 88.67% respectively. The VGG16 has shown the lowest results with 8.65% accuracy, 82.57% precision, 81.99% recall, and 81.59% F1 score.

## Results of the transfer learning models with LBP features

For the detection of colon and lung cancer from histopathological images, the transfer learning models are employed on LBP features. The results shown in Table 2 reveal that the CNN model and transfer learning models have shown significant improvement in results when trained on LBP features. VGG16 has shown improved results with LBP features as compared to the results obtained with the full feature set. ResNet, MobileNet and EfficientNetB4 have shown significant improvement of 7% in accuracy results. However, IncpetionResNetV2 outperformed other models with 99.98% accuracy, 99.99% precision, recall and F1 score. The class-wise F1 score accuracy of the proposed model is shown in Table 3.

## Results of MCC using both feature set

To provide an in-depth analysis of the four metrics utilized in this research work MCC is used. The results of both feature sets are shown in Table 4. Table 4 demonstrates the superiority of the proposed model using the LBP feature in all scenarios.

**Table 3  Class-wise results of the proposed InceptionResNetV2 model.**

| Class | F1-Accuracy |
|---|---|
| Lung benign tissue | 98.32% |
| Lung adenocarcinoma | 99.17% |
| Lung squamous cell carcinoma | 98.98% |
| Colon adenocarcinoma | 99.36% |
| Colon benign tissue | 99.99% |

**Table 4  Results of using both feature set.**

| Model | Using full feature set | Using LBP feature set |
|---|---|---|
| CNN | 0.84 | 0.89 |
| VGG16 | 0.80 | 0.85 |
| ResNET | 0.87 | 0.94 |
| EfficientNetB4 | 0.88 | 0.95 |
| MobileNet | 0.88 | 0.94 |
| Xception | 0.93 | 0.97 |
| InceptionResNetV2 | 0.96 | 0.99 |

**Table 5  K-fold cross-validation results.**

| Model | Accuracy | Precision | Recall | F1 score |
|---|---|---|---|---|
| fold-I | 99.68 | 99.98 | 99.93 | 99.67 |
| fold-II | 99.96 | 99.45 | 99.28 | 99.35 |
| fold-III | 99.59 | 99.48 | 99.48 | 99.48 |
| fold-IV | 99.78 | 99.35 | 99.69 | 98.59 |
| fold-V | 99.96 | 99.28 | 97.66 | 99.48 |
| **Average** | **99.88** | **99.45** | **99.42** | **99.76** |

**Notes.**
The averages of five fold cross validation results are shown in bold.

## Results of K-fold cross-validation

To demonstrate that the models performed optimally, K-fold cross-validation is utilised. Table 5 displays the results of a five-fold cross-validation, demonstrating that the proposed approach outperforms other models in terms of F1 score, accuracy, recall, and precision. This thorough validation technique underlines the superiority of the proposed method, highlighting its robustness and dependability in generating remarkable outcomes for colon cancer detection.

## Shapley additive explanations

SHAP (SHapley Additive exPlanations) (*Lundberg & Lee, 2017*) image data analysis is a strong approach for analysing the contributions of individual pixels or characteristics inside pictures to the predictions generated by machine learning models. It gives insights into the decision-making process of image classification algorithms, making it easier to comprehend and trust their conclusions. Deep learning models have demonstrated

exceptional performance in image categorization challenges. However, because of their intricacy, they are sometimes referred to as "black boxes." SHAP analysis assists in demystifying these models by offering a clear explanation with interpretation. SHAP analysis applies the Shapley values from the concept of cooperative game theory to images. The contribution of each pixel (feature) in the image to the model's output is represented by Shapley values. They calculate how much each pixel influences the forecast. Positive Shapley values suggest factors that influence the forecast upward, whereas negative values indicate features that influence the prediction downward.

SHAP analysis finds the top contributing features or pixels in an image for a specific prediction. This data assists in determining which regions or characteristics of a picture had the most effect on the model's judgement. SHAP includes visualisations to make the explanations more understandable. Making borders or overlays on images allows users to see which parts are crucial for the model's prediction. SHAP analysis of image-based data is useful in a variety of disciplines, including medical imaging used for disease detection and analysis. Image categorization systems that use SHAP analysis can be more robust, accurate, and ethical. It enables academics and practitioners to put their faith in and develop their models, resulting in better and more trustworthy image-based predictions.

This study employed the SHAP library to explain the predictions of a transfer learning model, especially for image data. The explanation variable contains the output of the explanation procedure offering insight into which sections of the image contributed to the model's predictions for the top five labels for lungs and colon cancer detection. The image plots created from the analysis provide insights into the model's behaviour and decision-making process. They visualize the impact that individual pixels have on the model's predictions for each image.

SHAP explanations show how individual features influence a specific instance's prediction. The sum of these contributions, along with the bias term, mirrors the model's initial prediction, capturing the forecast before the inverse link function is applied. Figure 3 presents the SHAP image plot for colon and lung cancer images from the LC25000 image dataset which offers a visual representation of the model's decision-making process, highlighting the influential pixels and contributing to the interpretability and transparency of the model's predictions for cancer classification. Pixels of darker regions contribute to a decrease in the model's output and pixels of lighter regions contribute to an increase in the model's output. High SHAP values in localized regions play a crucial role in the model's decision-making for colon or lung cancer classification. Visual differences in SHAP plots for colon and lung cancer images may reveal distinct patterns or features that the model relies on to differentiate between the two types of cancer.

## Discussion

In the presented results, two different sets of experiments were conducted to evaluate the performance of transfer learning models on histopathological images for the detection of colon and lung cancer. The experiments used a full feature set of the cancer image dataset and LBP features as inputs. The results from both sets of experiments are discussed below:

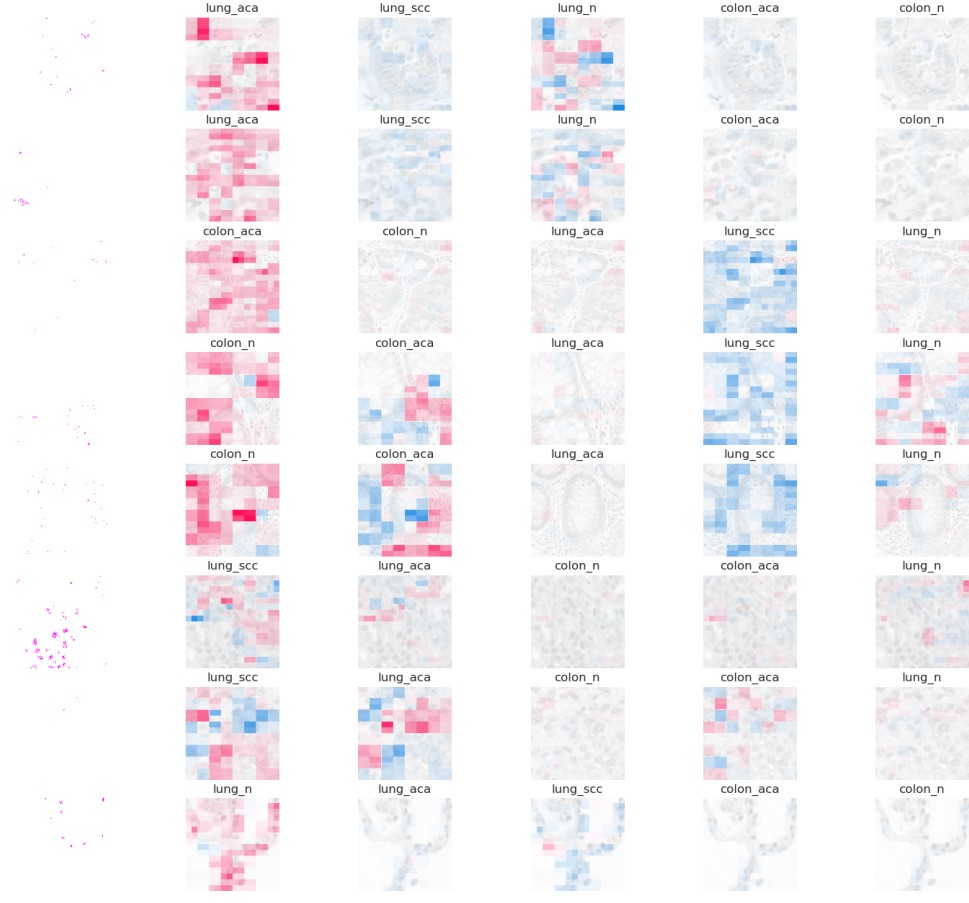

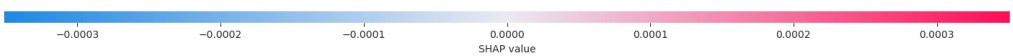

**Figure 3** **Graphical representation of SHAP feature importance.**

In the first set of experiments, the transfer learning models were applied to the full feature set of the cancer image dataset. The highest accuracy of 98.96% was achieved by the InceptionResNetV2 model, which also exhibited strong precision and F1 score as shown in Fig. 4. These results illustrate the varying performance of transfer learning models when applied to the complete feature set, with InceptionResNetV2 emerging as the top-performing model. In the second set of experiments, the transfer learning models were trained on LBP features, which are different feature representations. The results showed that most models achieved substantial improvements in accuracy when compared to using the full feature set as shown in Fig. 5. In particular, InceptionResNetV2 excelled with an exceptional accuracy of 99.98% along with high precision, recall, and F1 score. This
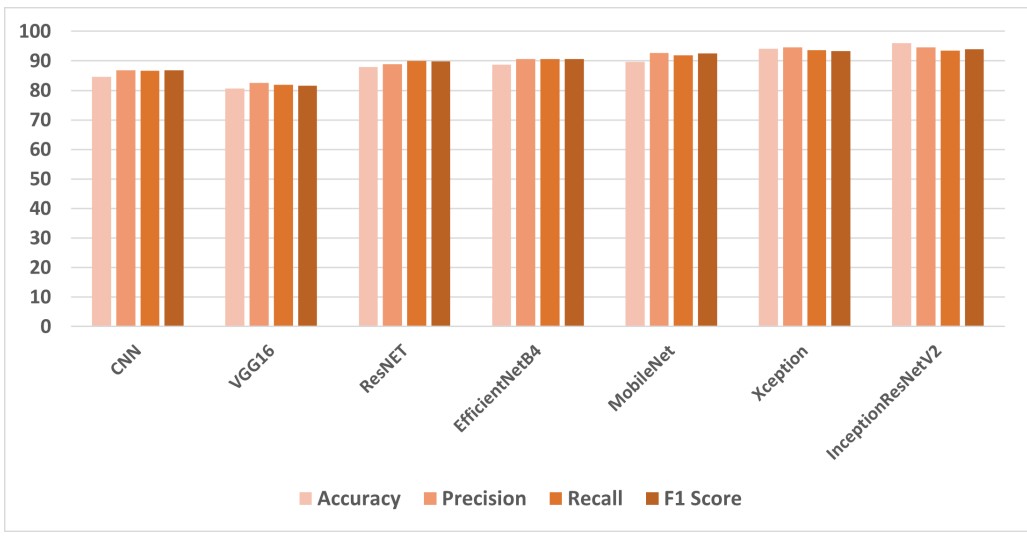

**Figure 4** Comparison of models using full features.

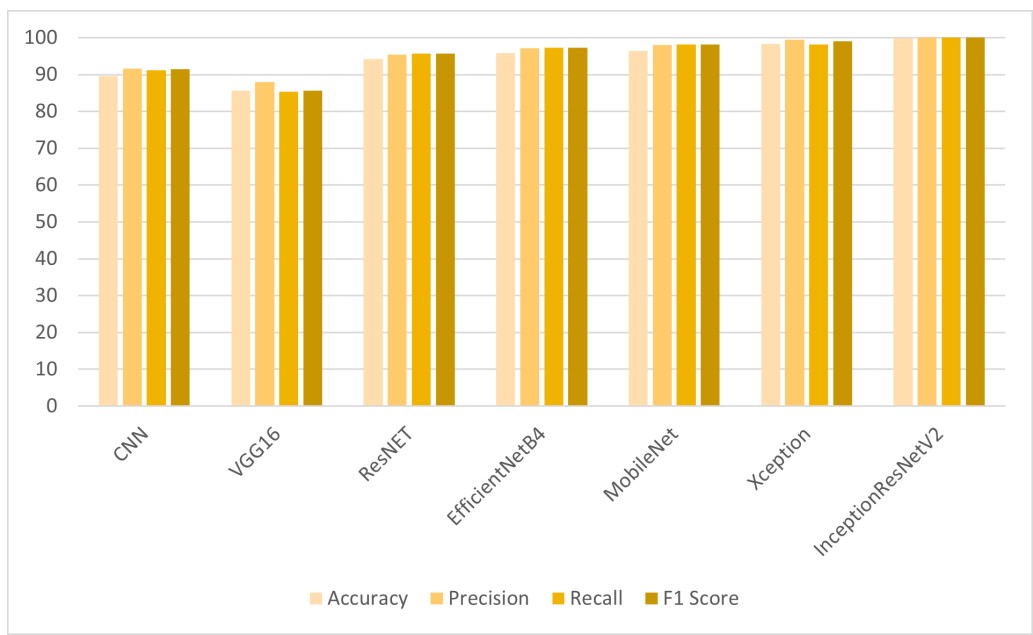

**Figure 5** Comparison of models using LBP features.

indicates the effectiveness of LBP features in enhancing the performance of the transfer learning models for cancer detection in histopathological images.

These results highlight the impact of feature selection on the performance of transfer learning models. While the full feature set provided strong results for some models, the use of LBP features led to significant improvements, especially for InceptionResNetV2, which achieved near-perfect accuracy. This demonstrates the importance of selecting the

right features for specific tasks, and it also underscores the effectiveness of transfer learning models for histopathological image analysis. InceptionResNetV2 is a deep CNN that has been pre-trained on a massive dataset of images, such as ImageNet. This means that it has already learned to recognize a wide range of generic image features, such as edges, textures, and shapes. When trained on a specific task, such as detecting cancer, InceptionResNetV2 can leverage this prior knowledge to better recognize the key patterns and characteristics that are indicative of the disease. This can lead to improved performance on the task compared to a model that has not been pre-trained. LBP features are widely recognized for their ability to effectively capture texture information in images. By employing LBP features as inputs to InceptionResNetV2, the model can adeptly utilize these texture descriptors to enhance its comprehension of histopathological images.

This synergistic integration of deep learning and texture-based features is likely a contributing factor to the model's remarkable performance. The study not only provides insights into the models' performance but also shows the potential for improving cancer detection using image analysis techniques. Additionally, the use of SHAP for model interpretation adds transparency to the predictions, helping to understand the models' decision-making process and the influence of individual features or pixels on the results.

The major reasons for misclassified images in the deep Inception-ResNet-v2 model with local binary patterns (LBP) features include (1) Variability in image characteristics: Images with varying lighting conditions, orientations, scales, or angles are not adequately represented by the extracted LBP features. This variability can lead to misclassifications as the model struggles to generalize across different image characteristics. (2) Limited discriminative power of LBP features: While LBP features are effective for capturing local texture patterns, they do not always provide sufficient discriminative power to differentiate between classes, especially in cases where classes exhibit subtle differences or overlap in feature space.

### Comparison with previous studies

Table 6 compares the proposed method's findings to some of the more well-known lung and colon cancer categorization approaches using the LC25000 dataset. As shown in the table, the proposed model exceeds the majority of cancer detection methods in terms of maximum classification accuracy. The proposed method, which uses the InceptionResNetV2 classifier, is also included for comparison.

*Hatuwal & Thapa (2020)* demonstrated a 97.2% accuracy utilizing a three-layer CNN model for the classification of colon and lung cancer images. Notably, their approach utilized 90% of the available data for training, a practice that introduces a potential risk of overfitting. Similarly, *Mangal, Chaurasia & Khajanchi (2020)* employed a three-layer CNN model, achieving classification accuracy while utilizing 80% of the dataset for training purposes. It is worth highlighting that a significant portion of the data was allocated to training, potentially influencing the model's generalization. Additionally, *Masood et al. (2018)* applied a three-layer CNN model, attaining a commendable accuracy of 96.33%. Their training strategy involved using 70% of the dataset over 500 epochs,

**Table 6   Comparison with state-of-the-art techniques using histopathological images.**

| Ref | Dataset | Proposed classifiers | Achieved accuracy |
|---|---|---|---|
| *Hatuwal & Thapa (2020)* | LC25000 (Lung And Colon Histopathological Image Dataset). | CNN | 97.2 |
| *Mangal, Chaurasia & Khajanchi (2020)* | LC25000 (Lung And Colon Histopathological Image Dataset). | CNN | 97.89 & 96.61 |
| *Masood et al. (2018)* | LC25000 (Lung And Colon Histopathological Image Dataset). | CNN | 96.33 |
| Proposed | LC25000 (Lung And Colon Histopathological Image Dataset). | InceptionResNetV2 | 99.89 |

indicating a different approach to data partitioning and model optimization compared to the aforementioned studies.

Notably, the proposed method achieved the highest accuracy of 99.89% on a dataset of lung and colon cancer images, surpassing the state-of-the-art techniques cited in the references. This table provides a concise summary of the performance of various classifiers, highlighting the effectiveness of the proposed classifier in achieving superior accuracy in the context of cancer image classification.

### *Significance of the proposed model*

To check the generalization and stability of the proposed model, this research work uses another independent dataset. This dataset is named "Wireless Capsule Endoscopy (WCE) Curated Colon Disease Dataset Deep Learning". The dataset is utilized by researchers to study gastrointestinal tract or colon diseases using images-based datasets. Because of the challenges associated with acquiring endoscopy images of gastrointestinal (GI) diseases, this study utilized existing datasets sourced from reputable repositories such as KVASIR (*Jha et al., 2020*) and ETIS-Larib Polyp DB (*Deeba, Bui & Wahid, 2020*). The KVASIR dataset originated from the Vestre Viken Health Trust in Norway and encompassed images from a diverse range of GI conditions obtained *via* wireless capsule endoscopy (WCE) procedures. Rigorous evaluation and data collection procedures were undertaken by medical professionals to ensure that all images in the dataset were accurately labeled, rendering them suitable for training and validation purposes. When trained and tested on this dataset, the proposed model gives a reliable accuracy score of 98.89%, 99.25% precision, 99.10% recall, and 99.18% F1 score. These reliable results on another dataset demonstrate the effectiveness of the proposed model in terms of medical-image-based analysis of different types of cancers.

## CONCLUSIONS

This study undertakes a comprehensive assessment of transfer learning models for the accurate detection of colon and lung cancer in histopathological images. The proposed approach is trained and validated using the LC25000 dataset. Two distinct feature sets are used to investigate the performance of these models: the whole feature set of the cancer picture dataset and LBP features. The relevance of feature selection in the context of medical

image analysis is clarified by the experiment findings. Most models improved significantly in accuracy when compared to the whole feature set, with InceptionResNetV2 standing out with an accuracy of 99.98%. This demonstrates the extraordinary synergy between LBP features and InceptionResNetV2 in improving cancer diagnosis in histopathology images.

This integration of deep learning and texture-based features stands as a pivotal factor behind InceptionResNetV2's remarkable performance. This research significantly shows how transfer learning models can be effectively leveraged for histopathological image analysis in cancer detection. The incorporation of SHAP for model interpretation enhances the transparency of the predictions, fostering a deeper understanding of the models' decision-making processes and the impact of individual features or pixels on the outcomes. These insights hold the potential to revolutionize cancer diagnosis in an era of more accurate and reliable image-based medical assessments. In the future, additional sets of characteristics derived from more histopathological images will be investigated in order to improve the performance.

### Funding
This study is supported via funding from Prince sattam bin Abdulaziz University project number (PSAU/2024/R/1445). The funders had no role in study design, data collection and analysis, decision to publish, or preparation of the manuscript.

### Grant Disclosures
The following grant information was disclosed by the author:
Prince sattam bin Abdulaziz University: PSAU/2024/R/1445.

### Competing Interests
The author declares there are no competing interests.

### Author Contributions
- Shtwai Alsubai conceived and designed the experiments, performed the experiments, analyzed the data, performed the computation work, prepared figures and/or tables, authored or reviewed drafts of the article, and approved the final draft.

### Data Availability
The code is available in the Supplemental File, GitHub and Zenodo:
- https://github.com/shtwai/Lung-ColonCancer
- shtwai. (2024). shtwai/Lung-ColonCancer: Preview files (preview). Zenodo. https://doi.org/10.5281/zenodo.10808988.
The data are available at Kaggle: https://www.kaggle.com/datasets/andrewmvd/lung-and-colon-cancer-histopathological-images.

## Supplemental Information

Supplemental information for this article can be found online at http://dx.doi.org/10.7717/peerj-cs.1996#supplemental-information.

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
