# Peer review of "Transfer learning based approach for lung and colon cancer detection using local binary pattern features and explainable artificial intelligence (AI) techniques"

_PeerJ Computer Science, doi:10.7717/peerj-cs.1996_

## Round 0.1 · original submission · Major Revisions

The reviewers have substantial concerns about this manuscript. The authors should provide point-to-point responses to address all the concerns and provide a revised manuscript with the revised parts being marked in different color.

**Language Note:** The review process has identified that the English language must be improved. PeerJ can provide language editing services - please contact us at [email protected] for pricing (be sure to provide your manuscript number and title). Alternatively, you should make your own arrangements to improve the language quality and provide details in your response letter. – PeerJ Staff

Reviewer 1 ·

Basic reporting

The research paper titled "Transfer learning based approach for Lung and Colon cancer detection using LBP features and Explainable AI techniques" introduces a study that aims to improve the diagnostic accuracy for lung and colon cancer identification using machine learning and deep learning approaches. The study proposes the use of the Inception-ResNetV2 model with strategically incorporated local binary patterns (LBP) features. The model is trained on histopathological images and achieves exceptional performance with 99.98% accuracy. The study also employs explainable AI techniques to provide transparency in decision-making processes. The document provides an overview of the importance of cancer diagnosis, the use of AI in cancer diagnosis, and previous studies in the field. It also outlines the contributions of the proposed approach and the structure of the paper. This topic is significant, there are several question need the author to address:

1.For model performance evaluation, could you provide the metric Matthews correlation coefficient (MCC) (Chicco & Jurman, 2020), which is a comprehensive and reasonable metric to balance four metrics used in your paper.

Chicco, D., Jurman, G. The advantages of the Matthews correlation coefficient (MCC) over F1 score and accuracy in binary classification evaluation. BMC Genomics 21, 6 (2020). https://doi.org/10.1186/s12864-019-6413-7

2.Of course, SHAP analysis can offer explainable details about features used in Deep learning model, yet, SHAP values is based on the aspect of the association between features and target in prediction or classification tasks, not causality. Honestly, SHAP analysis is helpful to understand the black box models. It is better pay more attention in its usage in practice, as the following paper mentioned. Please provide more comments on the use of SHAP with balanced reviews.

https://arxiv.org/pdf/2302.08160.pdf

Experimental design

Please see above.

Validity of the findings

Please see above.

Additional comments

Please see above.

Reviewer 2 ·

Basic reporting

no comment

Experimental design

Comment 1: The utilization of Local Binary Patterns (LBP) shows improved image representation capability compared to the full feature set. However, there are questions about how to regard the relationship between the full feature set and LBP features. Are these feature sets orthogonal, or do they overlap? In addition, it would be beneficial if the paper could explore the possibility of integrating both feature sets, perhaps through weighted balancing, to optimize information use.

Validity of the findings

Comment 2: The application of Explainable AI techniques is a commendable aspect of the study, expected to enhance the transparency and interpretability of the results. However, the paper falls short in fully demonstrating the advantages of SHAP. A more detailed explanation for Figure 3, possibly supplemented with pair-wise analyses that correlate feature importance with raw image input, would enrich the understanding of how SHAP feature importance score contributes to any findings.

Comment 3: The paper would benefit from presenting detailed accuracy metrics for each cancer category, enabling a more nuanced understanding of the model's performance across different types of lung and colon tissues.

Additional comments

Comment 4: The novelty of using LBP in lung and colon cancer tissue classification is questioned considering other similar methods, e.g. ‘Al-Jabbar, M., Alshahrani, M., Senan, E. M., & Ahmed, I. A. (2023). Histopathological Analysis for Detecting Lung and Colon Cancer Malignancies Using Hybrid Systems with Fused Features. Bioengineering, 10(3), 383’.

Comment 5: The rationale for selecting LBP over other texture descriptors should be more explicitly justified, especially in the context of findings from other studies like Kather, J. N., Weis, C. A., Bianconi, F., Melchers, S. M., Schad, L. R., Gaiser, T., ... & Zöllner, F. G. (2016). Multi-class texture analysis in colorectal cancer histology. Scientific reports, 6(1), 27988, which does not suggest LBP the optimal choice for texture analysis in cancer histology. Will other texture descriptors offer superior classification strategies for texture analysis in cancer histology?

Comment 6: A minor technical correction is suggested for line 167, where the image dimension should be stated as “768 by 768 pixels” for clarity.

Reviewer 3 ·

Basic reporting

1. The flow of the article is not very clear and concise. There is a lot of redundancy throughout the article, especially in the Related Work section, which plainly states the fact that it's difficult to diagnose lung and colon cancer without additional supporting evidence or statements. The overall takeaway from this section remains unclear to me.

2. In the proposed approach section, the author mentioned this study is to classify ovarian cancer in the very first line, which leads to confusion and distraction.

3. The author has discussed multiple prior model development works by others and referred to them just as 'The authors' in the Related Work section. It would be better to provide some inline references to avoid confusion.

4. Figure legends are not self-contained and Fig 3 is not even mentioned in the text.

Experimental design

1. The study objective is not well-defined and how it fills a current knowledge remains unclear to me. The author provides an overview of the different types of models and then goes straight into the model performance. There is no explanation or interpretation from the author on what is the advantages or limitations of using each different model.

2. The use of training data to tune the model was not described and whether the reported performance was based on training data or test data was never made clear to me.

3. There is no clear conclusion from the SHAP analysis.

4. In Table 4, the author reports the performance of the proposed approach is superior to the other state-of-the-art methods. It was not clear to me what's contained within the 'proposed approach', besides applying the available InceptionResNetV2 model developed by Google. Since the other state-of-the-art methods were applied to different datasets compared to the model, this is not a fair comparison and the author's conclusion on the superiority of the current model is biased.

Validity of the findings

The impact of the study was not clearly stated. There is no apparent novelty described in the article that can benefit the field. The conclusion derived from the article is not well-supported by the results.

Additional comments

This paper provides some good overview of the methods used in the field for diagnosing cancer using image recognition. However, the impact and novelty of the original research content within the article are very limited and not clearly stated at all. The comparison made between various methods is very shallow and discussion around the advantages and limitations of each method is absent. The use of SHAP to help with the interpretability of the model is plausible, but its takeaways and how that contributes to the overall research objective are not clear.

---

## Round 0.2 · Minor Revisions

There are only some remaining minor concerns that need to be addressed.

Reviewer 1 ·

Basic reporting

I have no further comments on this research paper and recommend its acceptance.

Experimental design

no

Validity of the findings

no

Additional comments

I am satisfied with the improvements of the draft and high quality of the code provided in the supplementary material.
I have no further comments on this research paper and recommend its acceptance.

Reviewer 2 ·

Basic reporting

Code should be shared publicly to enable other researchers to validate and build upon the work.

Experimental design

no comment

Validity of the findings

no comment

Reviewer 3 ·

Basic reporting

no comment

Experimental design

no comment

Validity of the findings

In the resubmission of the paper, the author addresses the previous comments and concerns. The model performance in characterizing different subtypes of lung and colon cancer is plausible. The author also attempted to elaborate more on the interpretation of SHAP results.
In the section on "comparison with previous studies", since the author is comparing the performance of the proposed model to other models trained on different sets of data, it would be great to see the performance of the proposed model on an independent dataset to further evaluate the generalization of the model.
It would also be beneficial if the author could comment on the misclassified images, although a few, to examine if there is any commonality.

---

## Round 0.3 · accepted · Accept

Minor concerns have been addressed and I would recommend accepting this manuscript.